# Embedded Vision System for Thermal Face Detection Using Deep Learning

**DOI:** 10.3390/s25103126

**Published:** 2025-05-15

**Authors:** Isidro Robledo-Vega, Scarllet Osuna-Tostado, Abraham Efraím Rodríguez-Mata, Carmen Leticia García-Mata, Pedro Rafael Acosta-Cano, Rogelio Enrique Baray-Arana

**Affiliations:** Tecnológico Nacional de México/Instituto Tecnológico de Chihuahua, Ave. Tecnológico 2909, Chihuahua 31310, Mexico; m23060252@chihuahua.tecnm.mx (S.O.-T.); abraham.rm@chihuahua.tecnm.mx (A.E.R.-M.); carmen.gm@chihuahua.tecnm.mx (C.L.G.-M.); pedro.ac@chihuahua.tecnm.mx (P.R.A.-C); rogelio.ba@chihuahua.tecnm.mx (R.E.B.-A.)

**Keywords:** face detection, thermal infrared sensors, deep learning, embedded vision systems

## Abstract

Face detection technology is essential for surveillance and security projects; however, algorithms designed to detect faces in color images often struggle in poor lighting conditions. In this paper, we describe the development of an embedded vision system designed to detect human faces by analyzing images captured with thermal infrared sensors, thereby overcoming the limitations imposed by varying illumination conditions. All variants of the Ultralytics YOLOv8 and YOLO11 models were trained on the Terravic Facial IR database and tested on the Charlotte-ThermalFace database; the YOLO11 model achieved slightly higher performance metrics. We compared the performance of two embedded system boards: the NVIDIA Jetson Orin Nano and the NVIDIA Jetson Xavier NX, while running the trained model in inference mode. The NVIDIA Jetson Orin Nano performed better in terms of inference time. The developed embedded vision system based on these platforms accurately detects faces in thermal images in real-time.

## 1. Introduction

Thermal infrared cameras use passive sensors to capture infrared radiation emitted by objects to build images. This class of sensors has been developed for many decades. Initially, they were used in the military as surveillance cameras and night vision tools, but their use was limited due to high costs. In recent years, prices have significantly decreased, allowing engineers to develop new applications, although these technologies remain inaccessible to everyone [1,2].

Thermal infrared cameras can be deployed to detect and recognize objects by applying algorithms based on Machine Learning (ML) or Deep Learning (DL). ML algorithms require relatively less computer power than DL algorithms. Still, the latter has higher precision and easier and faster processing because they use libraries implemented on GPU accelerators such as PyTorch, Tensorflow, OpenCV, and others [3].

Object detection is an essential task in computer vision. It involves determining the different classes of objects present in images, including humans. Due to the development of surveillance and security applications using computer vision techniques, the investigation has centered on human detection. This implies both the detection and segmentation of people [4]. Different approaches have been taken, some focusing on the complete body and others on faces. Recent advances in topological perception theory and persistent homology for topological feature extraction [5] can be used to build human body shape descriptors to improve human detection systems. Advances in the design of thermal imaging sensors or long-wave infrared (LWIR) imaging sensors have gained increasing attention for analyzing human and non-human images [6]. Human detection on thermal images can improve systems in the context of Autonomous Driving for pedestrian detection and trajectory planning [7,8], and occluded person re-identification (re-ID) methods [9].

Face detection is an important task in computer vision applications. It is the first step for face recognition and a crucial step since face recognition performance depends heavily on it [10]. Feature-based methods, in combination with ML algorithms for classification, like Viola–Jones [11], Histograms of Oriented Gradients (HOG) [12] combined with Support Vector Machines (SVMs) [13], and Multi-Block Local Binary Pattern (MB-LBP) [14], successfully detect faces in images in the visual spectrum and improved with the appearance of deep learning methods specially developed for this task, such as Multitask Cascaded Convolutional Networks (MTCNN) [15], cascade Convolutional Neural Networks (cascade-CNN) [16], Single Shot Scale-Invariant Face Detector (S3FD) [17], and FaceBoxes [18]. Some of these face-detection algorithms for images in the visual spectrum have been tested on thermal images. Three descriptors were evaluated in [19], i.e., Haar-like features, HOG, and LBP, using classifiers based on AdaBoost; the Max-Margin Object-Detection Algorithm (MM-ODA), joint with HOG and a DNN-based classifier, were also investigated. The results showed that face detection rates with thermal images are not comparable with those in the visual spectrum. In [20], an algorithm combining Multi-Block LBP with different descriptors, like Haar-like and HOG, was extended and used with an AdaBoost-based classifier to detect faces in thermal images. The results showed that combined features perform better than single-feature approaches. There are some algorithms developed specifically to detect faces in thermal images, like Eye Corner Detection (ED) [21] and Projection Profile Analysis (PPA) [22], but they were outperformed by algorithms designed to detect faces in visible images and applied to thermal images according to [23].

Recently, investigations on face detection in thermal images have used DL models. In [24], Silva et al. used transfer learning to train a general object detection model like YOLOv3 with their own created database to detect faces in thermal images in real time and reported its performance with a mAP50 of 99.7% and a mAP of 78.5%. In [10], Cao et al. trained the S3FD model with thermal face images, compared its performance against two classifiers based on Haar-like and HOG features and observed that S3FD performed better than traditional methods.

The novelty of the proposed system lies in training and comparing two state-of-the-art general-purpose object detection models, YOLOv8 and YOLO11, to detect faces in thermal images using two of the largest annotated thermal face databases available, the Terravic Facial IR database and the Charlotte-ThermaFace database, to subsequently execute the trained model in two embedded system boards, the NVIDIA Jetson Orin Nano and the NVIDIA Jetson Xavier NX, and finally, demonstrate that it is possible to detect faces in thermal images in real-time accurately. Compared to previous works [10] and [24], we utilized state-of-the-art deep learning models that were trained and tested on larger datasets, resulting in improved performance. The trained model was executed in inference mode on embedded system platforms for real-time face detection.

Traditionally, face detection systems have been developed using color cameras. However, certain disadvantages can complicate the detection process. Color cameras utilize sensors that capture light in the visible band of the electromagnetic spectrum; they are often adversely affected by poor lighting conditions, resulting in images that may be blurry, too dark, or excessively bright. To address these issues, thermal cameras equipped with infrared sensors have been utilized, as thermal images remain unaffected by inadequate lighting. However, the cost of thermal cameras is prohibitively high for conventional applications such as surveillance and biometric systems, and expenses increase significantly with higher resolution.

The motivation behind this research was to address the following questions: 1. Is it possible to accurately detect faces in thermal images in real time using an embedded vision system? 2. Can we train a state-of-the-art general-purpose object detection model to detect faces in thermal images? 3. Can we execute the trained model in inference mode on an embedded system board for real-time face detection? In this paper, we present the development of an embedded vision system specifically designed for real-time face detection in thermal images, intending to address these questions. This system utilizes low-cost thermal sensors connected to and operated from an embedded system board to capture thermal images. These images are analyzed using a deep learning model, which has been specifically trained for face detection. We trained and compared the performance of two state-of-the-art general-purpose object detection models: YOLOv8 and YOLO11. The Terravic Facial IR database was used for training, validation, and testing both models in all variants. Subsequently, we tested both models on a second database, the Charlotte-ThermalFace database. The experiments conducted, along with their results and analysis, are detailed in the subsequent sections of this paper. The trained model was executed in inference mode on two embedded system boards: the NVIDIA Jetson Orin Nano and the NVIDIA Jetson Xavier NX. We compared their performance based on inference time.

The main contributions of this work are:We conducted an experimental evaluation of two state-of-the-art general object detection models, YOLOv8 and YOLO11, trained to detect faces in thermal images. Two of the largest available thermal face databases were used: the Terravic Facial IR database was used for the training, validation, and testing, and the Charlotte-ThermalFace database was used to test the models further.We performed an experimental evaluation of two different embedded system boards, the NVIDIA Jetson Orin Nano and the NVIDIA Jetson Xavier NX, executing the trained model in inference mode. Both embedded platforms detected faces in thermal images accurately in real time.The developed embedded vision system can improve the face detection performance of surveillance and security systems, overcoming the limitations imposed by varying illumination conditions.

The remainder of this paper is structured as follows: Section 2 provides an overview of the hardware components utilized in constructing the embedded vision system, along with the software developed for analyzing thermal images for face detection. Section 3 presents the results of training, validation, and testing of two deep learning models, as well as the execution of the trained model in inference mode on two embedded system platforms. In Section 4, we discuss the outcomes of the conducted experiments. Conclusions are drawn in Section 5.

## 2. Materials and Methods

The development of an embedded vision system for thermal face detection using deep learning involves two critical stages: (1) selecting the appropriate hardware components and (2) developing the software necessary to analyze and detect faces in thermal images using deep learning models. The following sections provide detailed descriptions of both stages.

### 2.1. Hardware Components

The hardware components of the embedded vision system are detailed in the following sections.

#### 2.1.1. Thermal Sensors

In constructing the embedded vision system to detect faces in thermal images, we used a FLIR Lepton 3.5 thermal sensor; its technical specifications are summarized in Table 1. FLIR Lepton 3.5 sensors require a PureThermal 2 board offered by Groupgets. This board has a socket to insert the thermal sensor and the necessary firmware to read its data and transmit it to the embedded system board through a USB 2 interface. Figure 1 shows the PureThermal 2 board with the FLIR Lepton 3.5 thermal sensor.

#### 2.1.2. Embedded System Boards

The embedded system boards were selected based on two main capabilities: (1) image acquisition and processing for computer vision applications and (2) efficient execution of deep neural network (DNN) models in inference mode. Among the available options, the embedded system boards developed by NVIDIA stand out as the most viable option for constructing the embedded vision system for face detection in thermal images. These embedded system boards were designed specifically for computer vision applications with DNN acceleration. The ability to process information efficiently makes them ideal for projects requiring high-performance artificial intelligence and image processing tasks. A report comparing NVIDIA brand development cards at the hardware level was reviewed in [25], examining their main features.

Two embedded system boards were tested for constructing an embedded vision system for face detection in thermal images: (1) the NVIDIA Jetson Xavier NX Developer Kit and (2) the NVIDIA Jetson Orin Nano Developer Kit. These embedded system boards represent the best option regarding the cost-processing power ratio necessary for the system to operate. Figure 2 shows the NVIDIA Jetson Xavier NX and NVIDIA Jetson Orin Nano Developer Kits.

Even though these developer kits look similar, their architectures are different. Table 2 lists some important technical specifications of these embedded system boards, extracted from [25].

#### 2.1.3. Embedded Vision System

To build the embedded vision system, the casing for the NVIDIA Jetson Xavier NX and NVIDIA Jetson Orin Nano Developer Kits and supports for two PureThermal 2 boards were 3D printed; these supports are attached to the top of the board casing. The PureThermal 2 boards with the FLIR Lepton 3.5 sensors were placed in their supports and connected via USB cables to the ports of the developer kits. Figure 3 shows the assembled hardware of the embedded vision system using the NVIDIA Jetson Xavier NX Developer Kit.

### 2.2. Embedded Vision System Software Setup for Data Processing

The following subsections outline the initial setup of the embedded vision system and the installed software components. Next, we present the selected DL model and the thermal face database used for training, validating, and testing the DL model for face detection in thermal images. Finally, we explain how to execute the resulting DL model in inference mode on the embedded vision system.

#### 2.2.1. Initial Embedded System Board Configuration

The NVIDIA Jetson Xavier NX and the NVIDIA Jetson Orin Nano embedded system boards run a modified version of Ubuntu Linux, which is included in the software development kit JetPack SDK, provided by NVIDIA. The JetPack SDK provides a complete development environment for creating artificial intelligence (AI) applications. It includes Jetson Linux with a bootloader, a Linux kernel, an Ubuntu desktop environment, and a complete set of libraries for GPU computing acceleration, multimedia, graphics, and computer vision.

The JetPack SDK is installed on a microSD memory, which is inserted into the developer kit’s main board and used to boot the operating system. The JetPack SDK includes libraries such as TensorRT, cuDNN, CUDA, and OpenCV. For the NVIDIA Jetson Xavier NX Developer Kit, we installed JetPack SDK version 5.1.4, which includes Jetson Linux 35.6.0 Board Support Package (BSP) with Linux Kernel 5.10, an Ubuntu 20.04-based root file system, a UEFI-based bootloader, and OP-TEE as a Trusted Execution Environment. For the NVIDIA Jetson Orin Nano Developer Kit, we installed JetPack SDK version 6.2, which includes Jetson Linux 36.4.3 BSP with Linux Kernel 5.15, an Ubuntu 22.04-based root file system, a UEFI-based bootloader, and support for high-power Super Mode. AI performance is increased with Super Mode; the MAXN_SUPER is a power mode that allows the highest number of cores and clock frequency for the CPU and GPU engines.

TensorRT is a high-performance execution framework for deep learning neural network models in inference mode for object classification, segmentation, and detection. It is based on CUDA, NVIDIA’s parallel programming model, and allows for inference optimization of all deep learning models [26]. The NVIDIA Jetson Xavier NX embedded system board includes a deep learning inference optimizer and runtime (DLA) that offers low latency and high performance for AI applications. DLA functions are executed in a fixed-function accelerator engine implemented in hardware [27]. The cuDNN (CUDA Deep Neural Network) library provides high-performance primitives for deep learning models. It provides highly optimized implementations for standard routines such as forward and backward convolution, pooling, normalization, and activation layers [28].

The software for the embedded vision system for face detection in thermal images was developed in Python. The JetPack SDK version 5.1.4 installed on the NVIDIA Jetson Xavier NX Developer Kit includes Python 3.8 and OpenCV version 4.5.4. Other necessary libraries, such as Pytorch v2.1.0 and Torchvision v0.16.1, were also installed. These libraries were downloaded and compiled on the NVIDIA Jetson Xavier NX embedded system board; the installed versions correspond specifically to JetPack 5.1.4 and the CUDA v11.4 library. The JetPack SDK version 6.2 installed in the NVIDIA Jetson Orin Nano Developer Kit includes Python 3.10 and OpenCV version 4.11.0. We installed Pytorch v2.2.0 and Torchvision v0.17.0, corresponding to JetPack 6.2 and CUDA v12.6 library.

Image acquisition from the thermal sensors is performed using the OpenCV library functions. First, the capture object that makes the physical connection with the device connected to the USB port is initialized. Once the connection has been successfully made, image frames can be obtained from the thermal sensors.

#### 2.2.2. DNN Model Training, Validation, and Testing

The software for face detection is based on a pre-trained DNN model. The network model used is YOLO11, the latest version of Ultralytics YOLO (You Only Look Once) real-time object detectors, which feature improved accuracy, speed, and efficiency. Compared to previous versions of YOLO, YOLO11 introduces significant enhancements to its architecture and training methods. An in-depth review of its architectural structure, along with a comparison to previous versions, can be found in [29]. It features an improved backbone and neck architecture, enhancing feature extraction capabilities for more precise object detection and better performance on complex tasks. The YOLO11 model is available in five variants: YOLO11n (nano), YOLO11s (small), YOLO11m (medium), YOLO11l (large), and YOLO11x (extra-large). Each variant varies in architectural size, utilizing a different number of layers and trainable parameters; the choice of variant depends on the application’s complexity [30].

The YOLO11 model can be used for object detection with the parameters pre-trained on the MS COCO database, containing 80 different object types. It can identify the location and class of objects in a single image or video sequence. The output of an object detector is a set of bounding boxes that enclose the detected objects, along with class labels and confidence scores for each of them [31].

The YOLOv8 model in all its variants (n, s, m, l, and x) was trained, validated, and tested, its performance was compared to that of the YOLO11 model. YOLOv8 introduced architectural refinements to previous versions, including a refined CSPDarknet backbone, a C2f neck for improved feature fusion, and an anchor-free, decoupled head. In contrast, YOLO11 introduced additional architectural refinements, particularly focusing on optimizing the backbone, neck, and head structures to reduce the number of parameters while maintaining the anchor-free approach. YOLO11 incorporated the C3k2 block, SPPF (Spatial Pyramid Pooling—Fast), and C2PSA (Convolutional Block with Parallel Spatial Attention) components, which enhance the model’s ability to extract features and detect objects [32].

The YOLOv8 and YOLO11 models were trained to detect faces in thermal images. To accomplish this, it was necessary to search for available databases of facial thermal images online that included annotations, meaning they contain manually defined data for the bounding boxes enclosing the faces in each image from the database. The Terravic Facial Infrared Database [33] was utilized, which consists of 23,333 thermal images of faces belonging to 20 different male individuals captured indoors and outdoors, at various angles, some wearing glasses, mustaches, beards, and/or caps. All images are 320 × 240 pixels in size. The original annotations were converted from Pascal VOC format to YOLO format. The complete dataset was randomly divided into three subsets: (1) training (70%—16,333 images), (2) validation (20%—4667 images), and (3) testing (10%—2333 images). Figure 4 displays some images from the Terravic Facial Infrared database.

The Charlotte-ThermalFace database [34] was used as a second option for testing the trained models. This database contains 10,376 thermal images from 10 subjects, 5 male and 5 female, in four different room temperatures, 10 relative distances from the camera (1 m to 6.6 m), and 25 head positions. The image size varies with the distance from the camera: 465 × 350 pixels at 1 m, 300 × 225 pixels at 1.4 m, 295 × 220 pixels at 1.8 m, 250 × 185 pixels at 2.2 m, 200 × 150 pixels at 2.8 m, 180 × 140 pixels at 3.4 m, 150 × 130 pixels at 4 m, 120 × 85 pixels at 4.6 m, 100 × 80 pixels at 5.6 m, and 80 × 80 pixels at 6.6 m. The thermal face images were manually annotated with 72 or 43 facial landmarks. A CSV file containing these annotations was provided. Only 8143 images have annotations in the CSV file. We computed the bounding boxes in YOLO format from the annotated facial landmarks for each image by taking the minimum and maximum pixel coordinates in the horizontal and vertical axes. Since these bounding boxes were too tight to the faces, we increased their size by 30% to make them comparable with the bounding boxes of the Terravic Facial IR database. Figure 5 displays sample images from the Charlotte-ThermalFace database.

The YOLOv8 and YOLO11 models were trained on a workstation with a 13th-generation Intel Core i9 processor with 128 GB of DDR5 memory and an NVIDIA GeForce RTX3090Ti video card with 24 GB of GDDR6x memory. The workstation has installed the Windows 11 operating system, and the PyCharm development environment was used to create a virtual environment for the training projects for face detection in thermal images. The Torch v2.4.2 and Utralytics v8.3.25 libraries were installed in the virtual environment.

The YOLOv8 and YOLO11 models were trained in all their variants (n, s, m, l, and x) using the Terravic Facial Infrared database for 100 epochs with batches of 320 × 240 pixel thermal images. The use of GPU processing acceleration was explicitly specified. The batch size varies depending on the model variant and the available memory on the GPU card. The pre-defined training hyperparameters for the YOLOv8 and YOLO11 models were utilized. Table 3 displays the pre-defined hyperparameters for the training engine. The pre-trained weights on the MS-COCO database were used as initial weights, and we did not freeze any layer, changing the head of the model to classify only one class (thermal face).

The same initialization parameters were used for both models in all their variants. The Ultralytics package automatically downloads a file containing the pre-trained weights for the selected model variant in PyTorch format. The information about the structure and location of the database to be processed is stored in a YAML file. The pre-defined optimizer is AdamW with a learning rate (lr) of 0.002 and momentum of 0.9. Table 4 enumerates batch size, GPU memory used, number of layers, total number of parameters, gradients computed, GFLPOs performed during processing, training time per epoch, and validation time per epoch of the YOLOv8 and YOLO11 models in their different variants.

The batch size is adjusted for each model variant to utilize all available GPU memory and minimize processing time. Although the number of layers has increased, a reduction in the number of parameters can be observed when comparing the YOLOv8 model to the YOLO11 model in each variant. Figure 6 displays sample batches of thermal images from the training subset.

The Ultralytics package offers methods for data augmentation, including certain forms of ablation that involve removing portions of faces in the images used during training, as displayed in some images of Figure 6. The default data augmentation parameters were not modified, which include:Random hue, saturation, and value augmentations range in HSV space (hsv_h = 0.015, hsv_s = 0.7, hsv_v = 0.4),Maximum translation augmentation as fraction of image size (translate = 0.1),Random scaling augmentation range (scale = 0.5),Probability of horizontal image flip (fliplr = 0.5),Probability of using mosaic augmentation, which combines 4 images (mosaic = 1.0),Randomly erases regions of the image (erasing = 0.4),Randomly crop regions of the image (crop_fraction = 1.0),

Once the training and validation stages are finished, and the model weights are stored in a file in PyTorch format. The model achieved the best performance during training with this set of weights and is used in the test stage to detect thermal faces in a subset of the database whose images were not used for training or validation. All variants of the YOLO11 model were tested, and performance was measured in terms of the Intersection over Union (IoU) calculated between the annotated bounding box (BB1) and the predicted bounding box (BB2) using the equation:(1)$$IoU=\frac{area(BB1\;\cup \;BB2)}{area(BB1\;\cap \;BB2)}$$

The results of the training, validation, and testing stages are shown in the next section.

#### 2.2.3. DNN Model in Inference Mode

The trained model was prepared to be executed in inference mode in the embedded vision system to process thermal images. These images were captured using FLIR Lepton sensors mounted on PureThermal 2 boards, which are connected to the USB ports of the embedded system board. The model weights file, originally in PyTorch format, is exported to TensorRT format. For the NVIDIA Jetson Xavier NX embedded system board, TensorRT allows running parts of the model on the DLA processor and the rest on the GPU to accelerate the execution of the model in inference mode. The use of DLA is specified during the model conversion process. For the NVIDIA Jetson Orin Nano embedded system board, TensorRT executes the model using the GPU since its architecture does not include DLA processors. The embedded vision system, utilizing the NVIDIA Jetson Xavier NX and the NVIDIA Jetson Orin Nano Developer Kits, was tested for face detection in thermal images while executing the trained model in inference mode.

## 3. Results

### 3.1. DNN Models Training

The YOLOv8 and YOLO11 models were trained in their different variants for 100 epochs. The system processes 16,333 thermal images from the training subset, divided into batches, at each epoch. It computes the loss functions for bounding box location (box_loss), class prediction accuracy (cls_loss), and distribution of focal loss (dfl_loss), and the box precision and recall. It then processes 4667 thermal images from the validation subset, also divided into batches, computing the validation box, class, and dfl losses; the mean average precision calculated at an intersection over union (IoU) threshold of 0.5 (mAP50), and the average of the mean average precision calculated at varying IoU thresholds ranging from 0.5 to 0.95 (mAP50-95). The Ultralytics package computed the training and validation loss functions and the performance metrics. More information on performance metrics can be found in [35]. Training and validation loss function plots are displayed in Figure 7 for the YOLOv8 and YOLO11 models in their different variants.

Box and Class loss functions produced similar plots for the YOLOv8 and YOLO11 models across all their variants. The DFL loss function plots display different convergence values for two groups of mixed models and variants, with the YOLOv8n, YOLO11n, and YOLO11s models in the group with the lowest DFL loss function values. Figure 8 presents the performance metrics plots for box precision, box recall, mAP50, and mAP50-95 for both models across all their variants.

Precision and recall metrics reach a value of one early in the training process, indicating that the annotated thermal faces were detected and classified correctly in all training images. The mAP50 values approaching one indicate that most IoU values significantly exceed the 0.5 threshold. Furthermore, the mAP50-95 values are all above 0.9, suggesting minimal differences in size and location between the predicted and annotated bounding boxes.

The system saves the adjusted set of weights to a file named ‘last.pt’ at the end of each training epoch. It also monitors the best-performing set of weights and stores it in a separate file named ‘best.pt’.

### 3.2. DNN Models Validation

The YOLOv8 and YOLO11 models were validated in their different variants using the best set of weights. The validation results, including mAP50, mAP50-95, and inference time per image, are shown in Table 5.

We observe comparable validation performance between YOLOv8 and YOLO11 across all variants when utilizing the best sets of weights. Inference times are similar across the models; however, they increase significantly with medium, large, and extra-large model sizes.

Figure 9 displays samples of thermal images from the validation subset with (a) annotated bounding boxes and (b) predicted bounding boxes generated by the YOLO11s model. It is hard to find a difference in the size and location of the annotated and predicted bounding boxes from these images.

### 3.3. DNN Models Testing

The YOLOv8 and YOLO11 models were tested in their different variants using a subset of 2333 thermal images from the Terravic Facial IR database, not used during training or validation. Table 6 shows the testing stage results, including the average IoU between annotated and predicted bounding boxes, the number of samples with an IoU score below 0.8, and the inference time per image. Precision and Recall metrics reached 1 for all model variants.

The IoU performance metric is consistent across both models and their variants. An average IoU score above 0.93 indicates a minimal difference in size and location between the annotated and predicted bounding boxes. Figure 10 displays samples of thermal images from the test subset, with the annotated bounding boxes in red and the predicted bounding boxes in blue generated by the YOLO11s model.

Figure 11 displays images from the test subset in which the YOLO11s model predicted bounding boxes with IoU scores below 0.8, compared to the annotated bounding boxes. The IoU threshold of 0.8 was chosen because very few samples fell below this value.

In the first image of Figure 11, the annotated bounding box in red does not include the ear, while the predicted bounding box in blue includes it, resulting in an IoU score of 0.72606. In the second image, the annotated bounding box is smaller than the predicted bounding box, yielding an IoU score of 0.76506. Thermal face detection performs adequately in both cases, despite the smallest IoU values.

The Charlotte-ThermalFace database was used for testing to further assess the performance of the YOLOv8 and YOLO11 models across all their variants. Table 7 shows the test results on the Chalotte-ThermalFace database, including the average IoU between annotated and predicted bounding boxes, accuracy, and the inference time per image.

The Charlotte-ThermalFace database and the Terravic Facial IR database differ in four key aspects: (1) The varying distances of faces from the camera, which result in different image sizes in the Charlotte-ThermalFace database compared to the fixed image size in the Terravic Facial IR database. (2) The annotated data are different; the bounding boxes in the Terravic Facial IR database were defined manually to enclose the forehead, ears, and part of the neck, while the bounding boxes in the Charlotte-ThermalFace database were initially defined using the minimum and maximum coordinates of manually annotated facial landmarks. However, these bounding boxes were too tight around the facial landmarks, so we increased their size by 15% in all four directions to enclose larger portions of the face. Since this increment is uniform in all directions, the bounding box may be incorrectly enlarged when the head is rotated to one side. (3) Different number of head positions; in the Terravic Facial IR database, the subjects rotate their heads to both sides, generating no more than 10 different head positions, while in the Charlotte-ThermalFace database, the subjects rotate their heads to both sides, and up and down, resulting in 25 head positions. (4) The Terravic Facial IR database contains images from only male subjects, while the Charlotte-ThermalFace database contains images from five male and five female subjects.

These key aspects have produced differences in performance. When testing on the Terravic Facial IR database, both models detected faces in all the images of the test set, while testing on the Charlotte-ThermalFace database, both models made no predictions for some images, which we consider as False Positives. Since there are no False Negatives (objects detected as faces), the performance is measured with the Accuracy metric, as shown in Table 7, where the YOLO11 model has slightly higher accuracy scores than the YOLOv8 model, meaning that feature extraction in YOLO11 performs better. Figure 12 displays sample images where the model made no predictions. The annotated bounding boxes are shown in red.

Most of the non-detected faces occurred with female subjects rotating their heads to one side, and bending downwards or upwards. Some other cases of no predictions occurred when the male or female subjects were bending their heads sideways. These head postures were not observed in the images of the Terravic Facial IR database used for training.

The Average IoU scores shown in Table 7 indicate that YOLOv8 and YOLO11 models achieved similar performances in all their variants, with the YOLO11 model having slightly higher scores. The Average IoU scores below 0.63 indicate significant differences in the size and location between the annotated and predicted bounding boxes. These differences were caused by the method used to define the annotated bounding boxes for the Charlotte-ThermalFace database. Figure 13 displays sample images with the annotated bounding boxes in red and the predicted bounding boxes in blue.

The difference in size and location between the annotated bounding boxes in red and the predicted bounding boxes in blue is evident in the images of Figure 13, but it is also evident that the predicted bounding boxes enclose the thermal faces even in small-sized images.

### 3.4. DNN Model Executed in Inference Mode

We decided to implement the YOLO11s model in the embedded vision system, considering model size, performance metrics, and inference times. The file containing the model weights was converted from PyTorch format to TensorRT format to optimize the performance when executed in inference mode on the NVIDIA Jetson boards. For the NVIDIA Jetson Xavier NX embedded system board, DLA was set for model conversion, and power management was set to Mode 8—20 W—6 cores. For the NVIDIA Jetson Orin Nano embedded system board, the GPU was set for model conversion, and power management was set to Mode 2 MAXN SUPER. The Jetson_Clocks script was executed on both platforms to ensure the maximum clock frequency for the CPU and GPU engines. In the case of the NVIDIA Jetson Orin Nano Developer Kit, this resulted in a significant reduction in inference time.

The execution times of both platforms were compared while running the original PyTorch model, which was obtained after the training stage, and the converted TensorRT model. CPU time was measured for (1) pre-processing stage time, including image capture from a thermal sensor, color conversion, and resizing; (2) model inference time; and (3) post-processing stage time, including bounding box drawing. The results presented in Table 8 show the average CPU times recorded after capturing ten thermal images with detected faces.

Based on inference time, the NVIDIA Jetson Orin Nano embedded system board outperforms the NVIDIA Jetson Xavier NX embedded system board by more than half. Comparing the CPU times of the PyTorch model to the TensorRT models with DLA in the case of the Jetson Orin Nano and with GPU in the case of the Jetson Xavier NX, the difference is noticeable in terms of post-processing time. When analyzing the total CPU time, the NVIDIA Jetson Orin Nano embedded system board running the TensorRT model with GPU enabled is the better option, consuming 8.65 ms per image on average.

We measured the wall time required to process one full cycle, which includes pre-processing, inference, post-processing, and image display. We found an average duration of 125 milliseconds across all four experiments in Table 8. This result indicates that the embedded system boards must wait for the thermal sensor to capture the next frame, as the FLIR Lepton 3.5 thermal sensor has a maximum capture rate of nine frames per second. This frame rate enables the embedded vision system to capture and process more than one image per cycle. By adding a second thermal sensor to the embedded vision system, we observed that it could capture and process two thermal images within the same 125 milliseconds.

Thermal image samples captured and processed by the embedded vision system, utilizing the NVIDIA Jetson Xavier NX Developer Kit to execute the YOLO11s TensorRT model with DLA in prediction mode, are displayed in Figure 14. The execution times enabled the processing of thermal images captured from two FLIR Lepton 3.5 sensors. The model successfully predicted green bounding boxes around the detected faces. Both sensors were oriented in the same direction, capturing the same persons for demonstration purposes. For surveillance applications, the thermal sensors should be oriented to cover a broader area. Similar results were achieved by the embedded vision system using the NVIDIA Jetson Orin Nano Developer Kit to execute the YOLO11s TensorRT model with GPU in prediction mode.

## 4. Discussion

Analysis of the training results for the different variants of the YOLOv8 and the YOLO11 models allows us to observe that the performance improves with bigger models. However, the difference in performance between variants is insignificant compared to the difference in model size (number of stages, total number of parameters, and computed gradients). Validation performance is almost the same for both models in all their variants. However, inference time is an important factor in deciding on using the small or nano variants of the model since the target platform is an embedded system. In the training, validation, and test stages, it was observed that both models in all variants detected all the annotated thermal faces when processing the Terravic Facial IR database.

The Charlotte-ThermalFace database was used to test the trained models and further assess their performance. The Average IoU and the accuracy metric showed that the YOLO11 model performed slightly better than the YOLOv8 model. Important differences between the Terravic Facial IR database used for training and the Charlotte-ThermalFace database used for testing, such as image sizes, head posture variations, different methods to generate annotated bounding boxes, and subject gender balance, produced lower performance metrics, but the behavior of the trained models was stable, obtaining similar results for both models in all their variants.

The IoU metric was calculated to measure the performance in the testing stage. The YOLOv8 and the YOLO11 models in all their variants reached similar scores above 0.93 when testing on the Terravic Facial IR database, showing that predicted bounding boxes have small differences in size and location compared with the annotated bounding boxes.

The YOLO11s model had fewer predictions with IoU scores below 0.8 and competing inference times in the test stage. After analyzing all these factors, we decided to execute the YOLO11s model in the embedded vision system to process thermal images. Two embedded platforms were tested. Based on inference time, the NVIDIA Jetson Orin Nano outperformed the NVIDIA Jetson Xavier NX due to its more advanced architecture. The execution times in the embedded vision system using both platforms were measured. We observed that the system must wait for the thermal sensor to capture frames, which allows it to capture and process multiple frames per cycle. Subsequently, a second sensor was integrated into the embedded vision system without impacting the execution time.

The Terravic Facial IR Database was selected due to its extensive collection of images, surpassing other online thermal face databases. Images in this database were captured with subjects positioned at a consistent distance from the camera, featuring only one subject per image. The YOLOv8 and the YOLO11 models trained on this database successfully detected the annotated faces across all images during the training, validation, and testing phases. However, when the trained model was executed in the embedded vision system, we observed that, despite employing various data augmentation techniques for training, including translation, resizing, and flipping of the input images, the faces were detected only within a specific range of distances from the system, similar to those in the training database.

## 5. Conclusions

An embedded vision system to detect faces in thermal images was developed. It can acquire and process images from two thermal sensors to detect faces. A state-of-the-art DNN object detection model processes the thermal images and predicts bounding boxes enclosing the detected faces. The YOLOv8 and the YOLO11 models in all their variants (n, s, m, l, x) were trained, validated, and tested on the Terravic Facial IR Database, being able to detect all the annotated faces in the dataset with all variants at all stages. The IoU metric was used to evaluate the predictions made by the model during the test stage, and all model variants obtained average IoU scores above 0.93, with a small number of images having IoU scores below 0.8 when testing on the Terravic Facial IR database. The Charlotte-ThermalFace database was used to test the trained models to further assess their performance. Although performance was reduced due to differences between these databases, the behavior of the trained models was stable, providing very similar results in both models and their variants.

The YOLO11s model was chosen based on its size and inference time and was executed on the embedded vision system to process thermal images in real time. Two embedded platforms were tested for executing the YOLO11s model. The model was exported to TensorRT format with DLA support and executed on the NVIDIA Jetson Xavier NX embedded system board. Additionally, it was exported to TensorRT format with GPU support and executed on the NVIDIA Jetson Orin Nano embedded system board. The embedded vision system, tested on both platforms, successfully detected faces in images captured from two thermal sensors in real-time.

The design and construction of an embedded vision system to detect faces in thermal images in real-time using a state-of-the-art object detection model specifically trained for this task has been proposed. The purpose of developing the embedded vision system is to improve the performance of surveillance and security systems, overcoming the limitations imposed by varying illumination conditions. To enhance the developed system, it will be necessary to construct thermal face databases that include more variants that can be learned by DNN models, making them more robust.

The developed embedded vision system can detect faces in thermal images when the subjects are positioned at distances similar to those in the training images. Its performance degrades with images containing few face pixels, but performance can be enhanced by incorporating new training samples that include subjects at various distances and higher resolution thermal sensors. Future work should focus on exploring different resizing parameters for data augmentation during the training process.

## Figures and Tables

**Figure 1 sensors-25-03126-f001:**
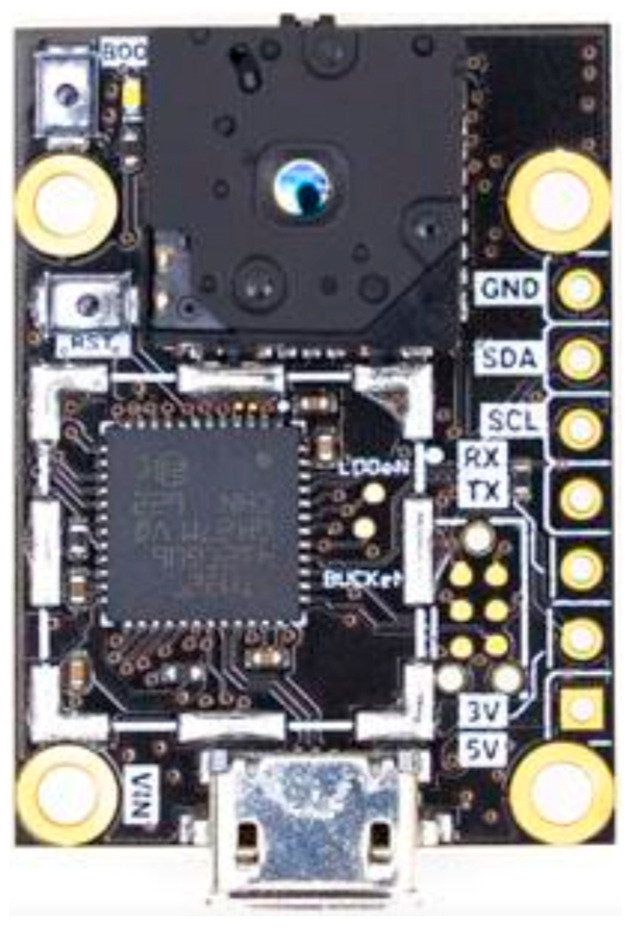
PureThermal 2 board with FLIR Lepton 3.5 thermal sensor.

**Figure 2 sensors-25-03126-f002:**
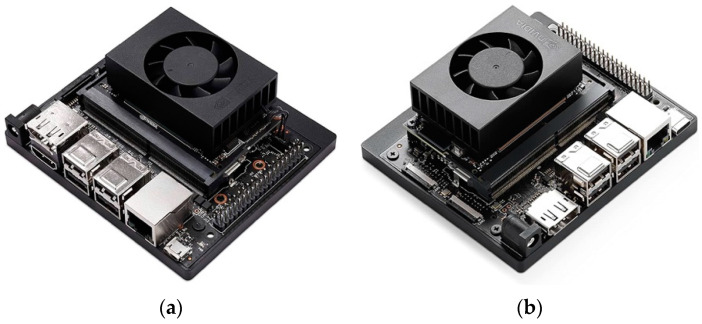
Embedded system boards: (**a**) NVIDIA Jetson Xavier NX Developer Kit; (**b**) NVIDIA Jetson Orin Nano Developer Kit.

**Figure 3 sensors-25-03126-f003:**
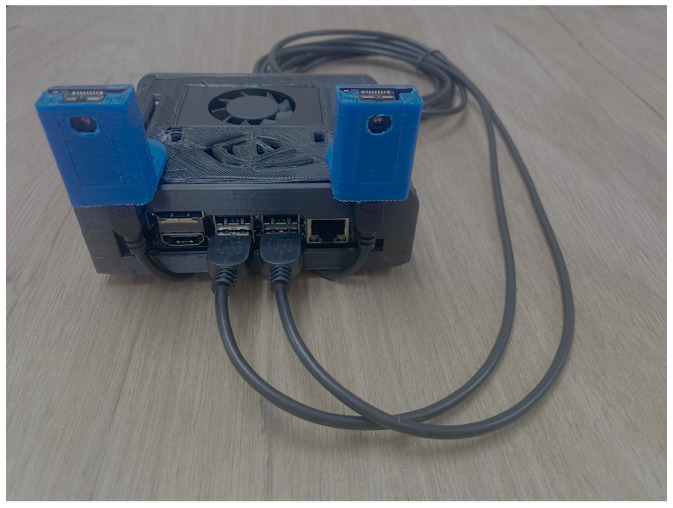
Embedded vision system assembled.

**Figure 4 sensors-25-03126-f004:**
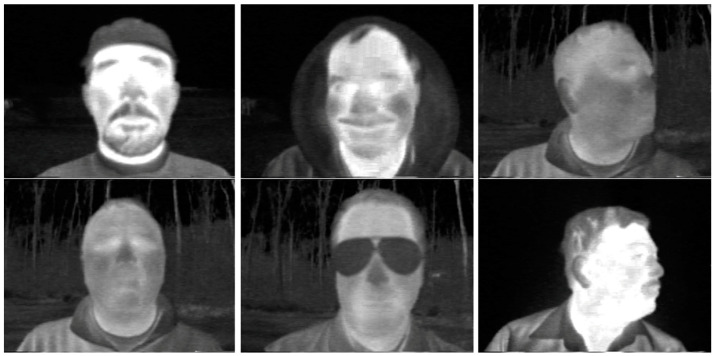
Sample images from the Terravic Facial Infrared database.

**Figure 5 sensors-25-03126-f005:**
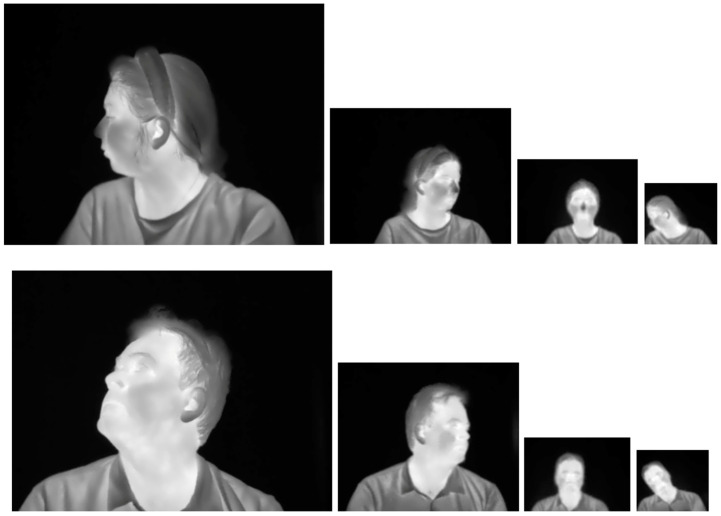
Sample images from the Charlotte-ThermalFace database.

**Figure 6 sensors-25-03126-f006:**
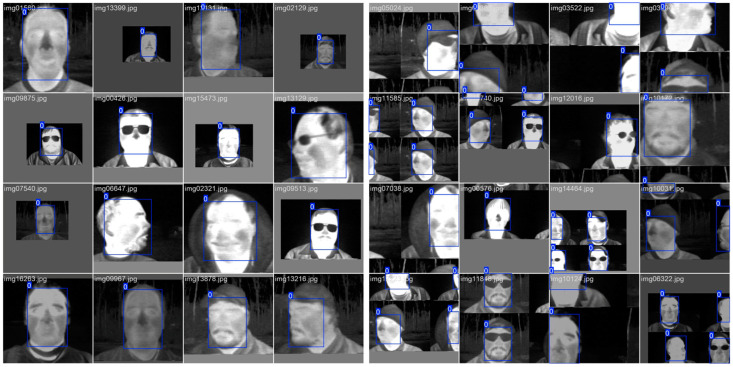
Sample batches of thermal images from the training set.

**Figure 7 sensors-25-03126-f007:**
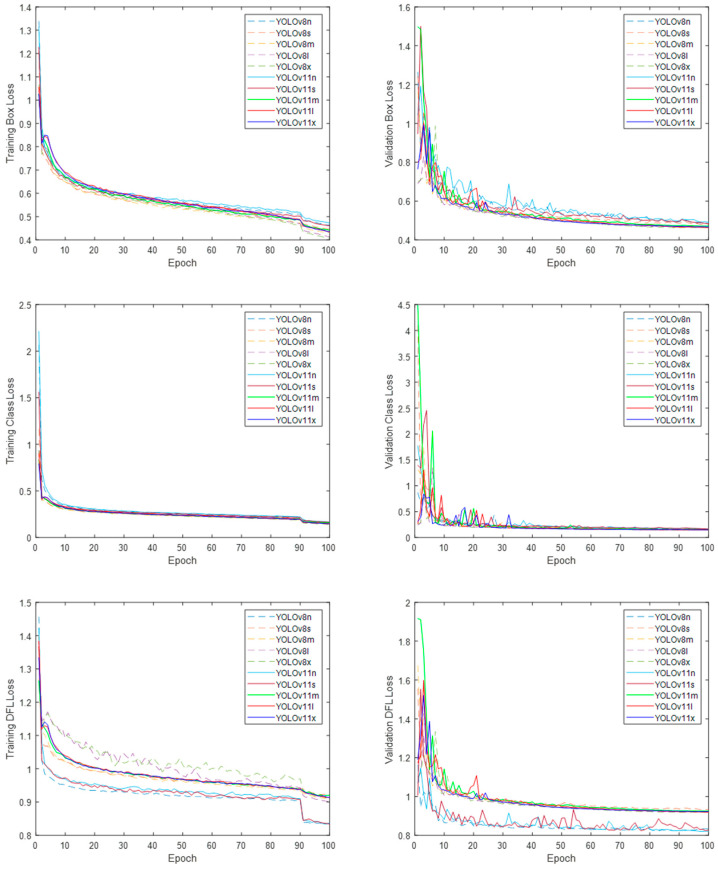
Training and validation loss functions plots for the YOLOv8 and YOLO11 models in their different variants.

**Figure 8 sensors-25-03126-f008:**
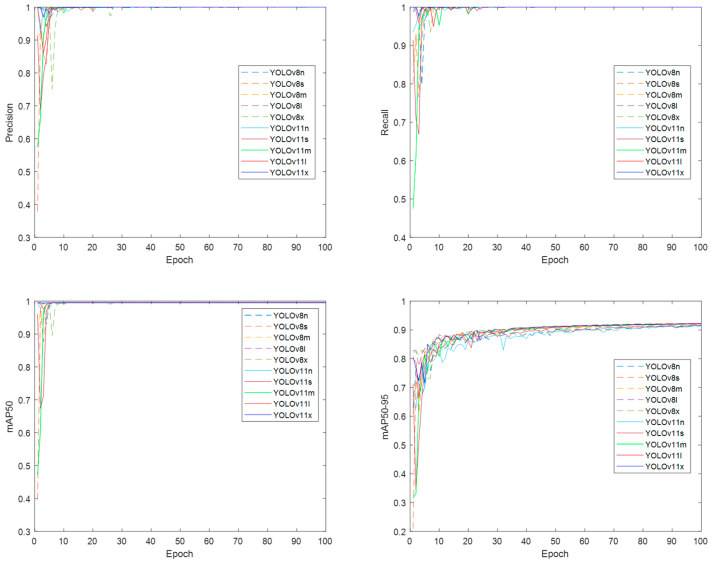
Training and validation performance metrics plots for the YOLOv8 and YOLO11 models in their different variants.

**Figure 9 sensors-25-03126-f009:**
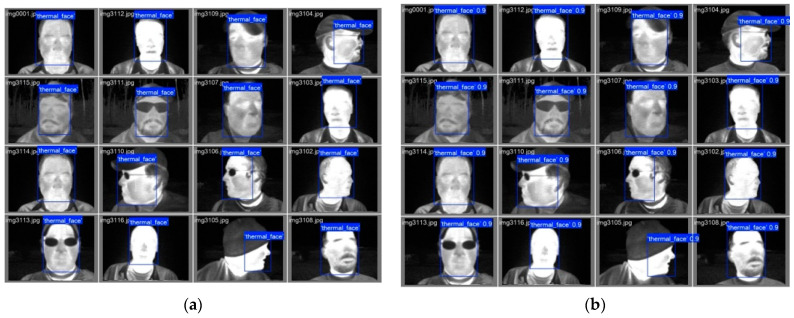
Validation sample images with (**a**) annotated bounding boxes and (**b**) predicted bounding boxes.

**Figure 10 sensors-25-03126-f010:**
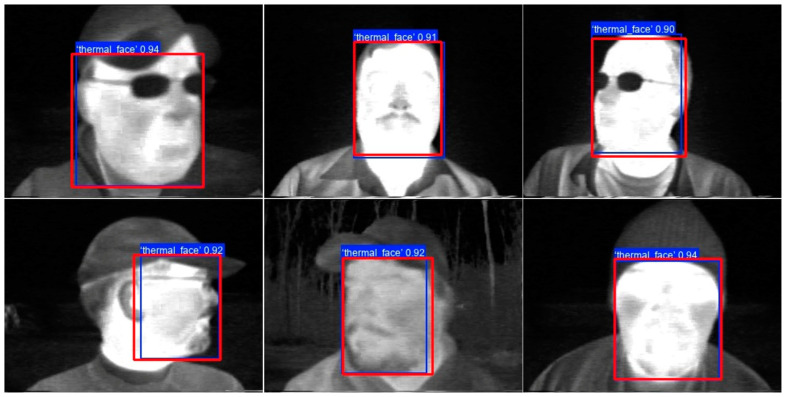
Test sample images with annotated bounding boxes in red and predicted bounding boxes in blue.

**Figure 11 sensors-25-03126-f011:**
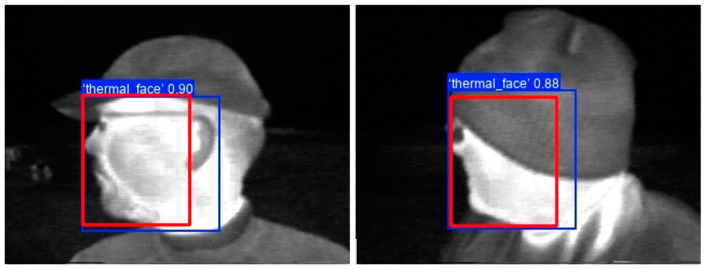
Test images with annotated bounding boxes in red and predicted bounding boxes in blue, with IoU scores below 0.8.

**Figure 12 sensors-25-03126-f012:**
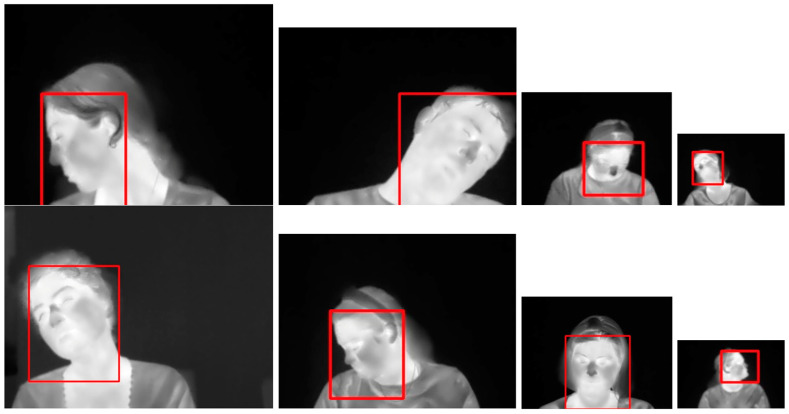
Sample images from the Charlotte-ThermalFace database where the trained models failed to detect faces.

**Figure 13 sensors-25-03126-f013:**
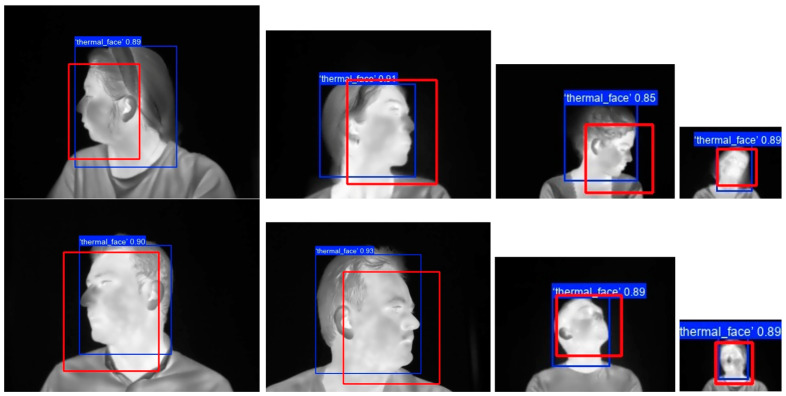
Sample images from the Charlotte-ThermalFace database with the annotated bounding boxes in red and the predicted bounding boxes in blue.

**Figure 14 sensors-25-03126-f014:**
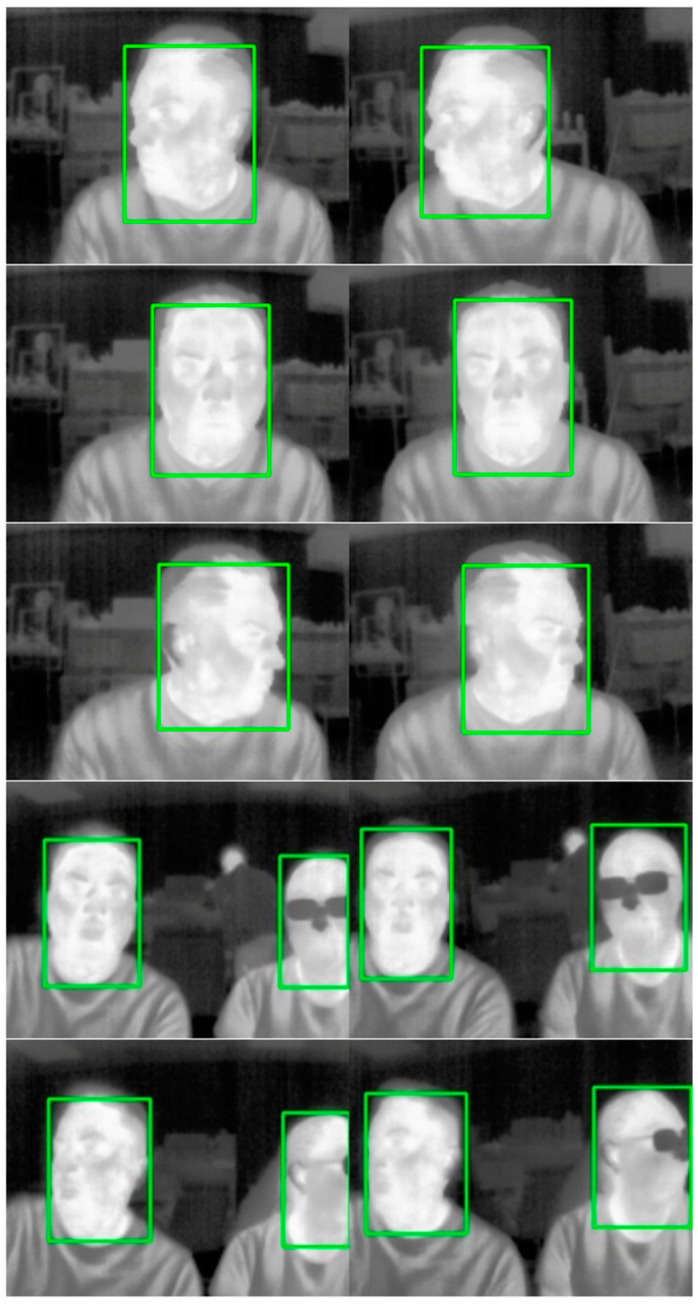
Face detection in thermal images from two sensors.

**Table 1 sensors-25-03126-t001:** Thermal sensor technical specifications.

Specifications	
Sensor Type	Uncooled Vox Microbolometer
Spectral Band	Longwave infrared 8 µm to 14 µm
Temperature Range	−10 °C to 140 °C
Sensitivity	<50 mK (0.050 °C)
Resolution	160 × 120
FPS	9 fps
Communication Interface	I2C (Control) SPI (Video) USB (with PureThermal 2)
FOV	57° Horizontal 71° Vertical

**Table 2 sensors-25-03126-t002:** Technical specifications of NVIDIA Jetson Xavier NX and NVIDIA Jetson Orin Nano Developer Kits.

Specifications		
	NVIDIA Jetson Xavier NX	NVIDIA Jetson Orin Nano
AI Performance	21 TOPS	67 TOPS
GPU	384-core NVIDIA Volta™ architecture GPU with 48 Tensor Cores	1024-core NVIDIA Ampere architecture GPU with 32 Tensor Cores
GPU Max Frequency	1020 MHz	1100 MHz
CPU	6-core Arm^®^ Cortex^®^-A78AE v8.2 64-bit CPU 1.5 MB L2 + 4 MB L3	6-core NVIDIA Carmel Arm^®^v8.2 64-bit CPU 6 MB L2 + 4 MB L3
CPU Max Frequency	1.7 GHz	1.9 GHz
Memory	8 GB 128-bit LPDDR5 102 GB/s	8 GB 128-bit LPDDR4x 59.7 GB/s

**Table 3 sensors-25-03126-t003:** YOLO11 model training hyperparameters.

Hyperparameter	Value
Learning Rate	0.01
Momentum	0.937
Weight Decay	0.0005
Warmup Epochs	3
Warmup Momentum	0.8
Warmup Bias Learning Rate	0.1

**Table 4 sensors-25-03126-t004:** Training parameters summary for YOLOv8 and YOLO11 in their different variants.

Model	Batch Size	GPU Memory	No. of Layers	Total Params	Gradients Computed	GFLOPS	Train Time/EPOCH	Val Time/EPOCH
YOLOv8n	640	19.7 Gb	225	3,011,043	3,011,027	8.2	12 s	6 s
YOLO v8s	384	21 Gb	225	11,135,987	11,135,971	28.6	20 s	7 s
YOLOv8m	216	21.4 Gb	295	25,856,899	25,856,883	79.1	39 s	9 s
YOLOv8l	148	22.4 Gb	365	43,630,611	43,630,595	165.4	60 s	13 s
YOLOv8x	120	22.1 Gb	365	68,153,571	68,153,555	258.1	94 s	20 s
YOLO11n	640	21.3 Gb	319	2,590,035	2,590,019	6.4	12 s	6 s
YOLO11s	384	23.5 Gb	319	9,428,179	9,428,163	21.5	19 s	7 s
YOLO11m	176	22.4 Gb	409	20,053,779	20,053,763	68.2	40 s	8 s
YOLO11l	128	21.1 Gb	631	25,311,251	25,311,235	87.3	53 s	9 s
YOLO11x	90	22.7 Gb	631	56,874,931	56,874,915	195.4	90 s	12 s

**Table 5 sensors-25-03126-t005:** Validation results for the YOLOv8 and YOLO11 models in their different variants.

Model	mAP50	mAP50-95	Inference Time
YOLOv8n	0.995	0.915	0.2 ms
YOLOv8s	0.995	0.919	0.2 ms
YOLOv8m	0.995	0.921	1.3 ms
YOLOv8l	0.995	0.922	1.93 ms
YOLOv8x	0.995	0.923	2.9 ms
YOLO11n	0.995	0.914	0.2 ms
YOLO11s	0.995	0.916	0.2 ms
YOLO11m	0.995	0.921	0.5 ms
YOLO11l	0.995	0.922	0.7 ms
YOLO11x	0.995	0.922	0.7 ms

**Table 6 sensors-25-03126-t006:** Test results for the YOLOv8 and YOLO11 models in their different variants.

Model	Average IoU	Samples with IoU < 0.8	Inference Time
YOLOv8n	0.93430	4	3.0 ms
YOLOv8s	0.93542	4	3.8 ms
YOLOv8m	0.93659	3	4.2 ms
YOLOv8l	0.93718	5	5.3 ms
YOLOv8x	0.937264	4	5.9 ms
YOLO11n	0.93338	5	3.6 ms
YOLO11s	0.93460	2	4.2 ms
YOLO11m	0.93566	5	4.5 ms
YOLO11l	0.93705	3	7.8 ms
YOLO11x	0.93639	3	8.4 ms

**Table 7 sensors-25-03126-t007:** Test results on the Charlotte-ThermalFace database.

Model	Average IoU	Samples with IoU < 0.8	Inference Time
YOLOv8n	0.628672	0.981206	2 ms
YOLOv8s	0.628224	0.920296	4.3 ms
YOLOv8m	0.616899	0.960196	20.3 ms
YOLOv8l	0.617041	0.954895	40.4 ms
YOLOv8x	0.624454	0.975424	71.1 ms
YOLO11n	0.628049	0.979375	1.9 ms
YOLO11s	0.639260	0.970605	3.3 ms
YOLO11m	0.624533	0.959811	31.9 ms
YOLO11l	0.623969	0.991326	41.3 ms
YOLO11x	0.623335	0.977158	81.7 ms

**Table 8 sensors-25-03126-t008:** Embedded vision system execution times.

CPU Time (Milliseconds)				
	Jetson Xavier NX		Jetson Orin Nano	
	PyTorch	TensorRT-DLA	PyTorch	TensorRT-GPU
Pre-processing	1.0	1.0	0.78	0.78
Inference	0.46	0.44	0.19	0.17
Post-processing	40	13	26	7.7

## Data Availability

The datasets used in this study are publicly available: the Terravic Facial IR database is available at http://vcipl-okstate.org/pbvs/bench/Data/04/download.html, accessed on 2 May 2025, and the Charlotte-ThermalFace database is available at https://github.com/TeCSAR-UNCC/UNCC-ThermalFace, accessed on 2 May 2025.

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
