# Peer review of "Embedded Vision System for Thermal Face Detection Using Deep Learning"

_sensors, 2025, doi:10.3390/s25103126_

Round 1
Reviewer 1 Report
Comments and Suggestions for Authors
- The motivation behind the work, the highlights of the proposed method compared to existing methods, and the significance of the approach are not clearly stated in the abstract and introduction. It is important to clarify the problems that the method addresses.
- From the main contribution points given in the introduction section, the novelty of the paper is not clear. According to the description, it seems that the authors have investigated and optimized some aspects that already exist in the SOTA (YOLO11 and NVIDIA Jetson Orin Nano).
- From the experimental results, it appears that the authors conducted their experiments solely on the Terravic Facial IR Database. This, however, is insufficient to demonstrate the robustness of the proposed method.
- Quantitative results would be necessary to provide more substantial support for the experimental results.
- What are the disadvantages of the proposed method?
The English expression of the full text needs polishing.
Author Response
Thank you for your comments to improve our paper. We addressed your observations as follows:
- The motivation behind the work, the highlights of the proposed method compared to existing methods, and the significance of the approach are not clearly stated in the abstract and introduction. It is important to clarify the problems that the method addresses.
Thank you for your observations. The abstract and introduction sections were modified to address your comments.
- From the main contribution points given in the introduction section,
the novelty of the paper is not clear. According to the description, it seems that the authors have investigated and optimized some aspects that already exist in the SOTA (YOLO11 and NVIDIA Jetson Orin Nano).
Thank you for your comment. A paragraph was added in the introduction section [page 2, line 74] to clarify the novel contributions of the developed system.
- From the experimental results, it appears that the authors conducted their experiments solely on the Terravic Facial IR Database. This, however, is insufficient to demonstrate the robustness of the proposed method.
Agree. We performed new experiments testing the models on the Charlotte-ThermalFace database and included an analysis of the results.
- Quantitative results would be necessary to provide more substantial support for the experimental results.
Thank you for your comment. We included experiments with the YOLOv8 model and quantitatively compared the performance with that of YOLO11.
- What are the disadvantages of the proposed method?
The main disadvantage of our system will be the low resolution of thermal sensors. Failure modes occur frequently with low-resolution images and a small number of face pixels. We modified the last paragraph of the conclusions section to explain it.
Reviewer 2 Report
Comments and Suggestions for Authors
Upon careful examination of your manuscript, I recognize its substantial potential and relevance to the field. Nevertheless, in order to fully realize its impact, several areas require improvement. Below, I outline the key aspects that would benefit from further revision:
- The abstract contains overly dense information. It can be simplified and should clearly highlight the core contributions of the paper, such as the specific techniques, models, and results used. I recommend reducing the background discussion of existing research to some extent, and instead, emphasizing the novelty and experimental results of the paper.
- In the inference mode, the model is optimized using TensorRT and executed on different hardware platforms. It is recommended to further evaluate the impact of the optimization process on performance, such as the effects of TensorRT conversion and DLA/GPU processing, as well as whether the performance differences between different hardware platforms are significant enough to support decisions regarding model selection and system design.
- Although the Terravic facial infrared database includes a variety of thermal images, there is a lack of discussion regarding the limitations of the dataset, such as whether the images in the database encompass various lighting conditions, backgrounds, and pose variations in different environments. Further analysis of the relationship between the dataset's diversity and the model's generalization ability would enhance the credibility of the study.
- To enhance the readability of the paper, please present the key parameters from Figure 5 in a three-line table.
- In line 250 of the paper, the authors mention: 'The YOLO11s model is shown in Figure 6.' From an English grammatical perspective, when referring to 'The YOLO11s model is shown in Figure 6,' it typically implies that the figure should display the architecture of the YOLO11s model, not some result images. This expression seems problematic.Additionally, the sudden introduction of the YOLO11s model, rather than other model variants, disrupts the logical flow of the paper.
- The paper mentions several evaluation metrics, such as mAP50, IoU, precision, and recall, but the specific methods for calculating and applying these metrics in different experiments are not clearly defined. It is recommended that the authors provide further clarification on the calculation process and application of these metrics in each experiment. Additionally, regarding the number of samples with an IoU score below 0.8, the authors should offer more background information and explain the significance and impact of this in the context of the experiments.
- The paper mentions that although all model variants successfully detected faces in thermal images during both training and testing, predictions with IoU scores below 0.8 were still considered acceptable. It is recommended to further analyze these cases with low IoU scores, particularly when faces are at extreme angles or farther from the sensor. Is the model's performance stable in such cases? Are there specific scenarios where the performance significantly declines?
- Although the performance of the YOLO11 model is presented in this paper, it is recommended that the authors include comparative experiments with traditional methods (such as HOG+SVM, Faster R-CNN, etc.) or other advanced models (such as YOLOv8, YOLOv9, etc.). By comparing the YOLO11 model with these alternatives, its advantages can be highlighted, and its innovation and practicality can be clearly demonstrated.
- It is recommended to include ablation experiments to validate the actual contribution of the improved backbone/neck structure to the feature extraction from thermal imaging.
- To help readers better understand the role and advantages of the various components of the embedded vision system for thermal face detection, it is recommended that the authors analyze the experimental data in Section 3 and place it immediately after the corresponding performance tables, rather than consolidating all the information in Section 4.
- The embedded vision system for thermal face detection has only been validated and tested on the Terravic Facial IR Database. Does it have practical applicability in an engineering context? The authors are encouraged to validate this through experiments and provide clarification.
- The conclusion section reiterates the authors' contributions. It is recommended that the authors simplify the language and focus on addressing what method was proposed, why it was proposed, what problems it solves, and the potential future developments in this area.
Upon careful examination of your manuscript, I recognize its substantial potential and relevance to the field. Nevertheless, in order to fully realize its impact, several areas require improvement. Below, I outline the key aspects that would benefit from further revision:
- The abstract contains overly dense information. It can be simplified and should clearly highlight the core contributions of the paper, such as the specific techniques, models, and results used. I recommend reducing the background discussion of existing research to some extent, and instead, emphasizing the novelty and experimental results of the paper.
- In the inference mode, the model is optimized using TensorRT and executed on different hardware platforms. It is recommended to further evaluate the impact of the optimization process on performance, such as the effects of TensorRT conversion and DLA/GPU processing, as well as whether the performance differences between different hardware platforms are significant enough to support decisions regarding model selection and system design.
- Although the Terravic facial infrared database includes a variety of thermal images, there is a lack of discussion regarding the limitations of the dataset, such as whether the images in the database encompass various lighting conditions, backgrounds, and pose variations in different environments. Further analysis of the relationship between the dataset's diversity and the model's generalization ability would enhance the credibility of the study.
- To enhance the readability of the paper, please present the key parameters from Figure 5 in a three-line table.
- In line 250 of the paper, the authors mention: 'The YOLO11s model is shown in Figure 6.' From an English grammatical perspective, when referring to 'The YOLO11s model is shown in Figure 6,' it typically implies that the figure should display the architecture of the YOLO11s model, not some result images. This expression seems problematic.Additionally, the sudden introduction of the YOLO11s model, rather than other model variants, disrupts the logical flow of the paper.
- The paper mentions several evaluation metrics, such as mAP50, IoU, precision, and recall, but the specific methods for calculating and applying these metrics in different experiments are not clearly defined. It is recommended that the authors provide further clarification on the calculation process and application of these metrics in each experiment. Additionally, regarding the number of samples with an IoU score below 0.8, the authors should offer more background information and explain the significance and impact of this in the context of the experiments.
- The paper mentions that although all model variants successfully detected faces in thermal images during both training and testing, predictions with IoU scores below 0.8 were still considered acceptable. It is recommended to further analyze these cases with low IoU scores, particularly when faces are at extreme angles or farther from the sensor. Is the model's performance stable in such cases? Are there specific scenarios where the performance significantly declines?
- Although the performance of the YOLO11 model is presented in this paper, it is recommended that the authors include comparative experiments with traditional methods (such as HOG+SVM, Faster R-CNN, etc.) or other advanced models (such as YOLOv8, YOLOv9, etc.). By comparing the YOLO11 model with these alternatives, its advantages can be highlighted, and its innovation and practicality can be clearly demonstrated.
- It is recommended to include ablation experiments to validate the actual contribution of the improved backbone/neck structure to the feature extraction from thermal imaging.
- To help readers better understand the role and advantages of the various components of the embedded vision system for thermal face detection, it is recommended that the authors analyze the experimental data in Section 3 and place it immediately after the corresponding performance tables, rather than consolidating all the information in Section 4.
- The embedded vision system for thermal face detection has only been validated and tested on the Terravic Facial IR Database. Does it have practical applicability in an engineering context? The authors are encouraged to validate this through experiments and provide clarification.
- The conclusion section reiterates the authors' contributions. It is recommended that the authors simplify the language and focus on addressing what method was proposed, why it was proposed, what problems it solves, and the potential future developments in this area.
Author Response
Thank you for your comments to improve our paper. We addressed your observations as follows:
1. The abstract contains overly dense information. It can be simplified and should clearly highlight the core contributions of the paper, such as the specific techniques, models, and results used. I recommend reducing the background discussion of existing research to some extent, and instead, emphasizing the novelty and experimental results of the paper.
Thank you for your recommendations. We modified the abstract to reflect your comments.
2. In the inference mode, the model is optimized using TensorRT and executed on different hardware platforms. It is recommended to further evaluate the impact of the optimization process on performance, such as the effects of TensorRT conversion and DLA/GPU processing, as well as whether the performance differences between different hardware platforms are significant enough to support decisions regarding model selection and system design.
Agree. We included a paragraph after Table 8 with the analysis of the performance of PyTorch and TensorRT models running in inference mode on the two embedded system platforms.
3. Although the Terravic facial infrared database includes a variety of thermal images, there is a lack of discussion regarding the limitations of the dataset, such as whether the images in the database encompass various lighting conditions, backgrounds, and pose variations in different environments. Further analysis of the relationship between the dataset's diversity and the model's generalization ability would enhance the credibility of the study.
Thank you for your comment. More details of the Terravic Facial IR database were included. We included new experiments to test the trained models on the Charlotte-ThermalFace database [page 8, line 274]. We also added a comparison between these two databases [page 14, line 442].
4. To enhance the readability of the paper, please present the key parameters from Figure 5 in a three-line table.
Thank you for pointing this out. Figure 5 was removed. Important training hyperparameters are now presented in Table 3.
5. In line 250 of the paper, the authors mention: 'The YOLO11s model is shown in Figure 6.' From an English grammatical perspective, when referring to 'The YOLO11s model is shown in Figure 6,' it typically implies that the figure should display the architecture of the YOLO11s model, not some result images. This expression seems problematic. Additionally, the sudden introduction of the YOLO11s model, rather than other model variants, disrupts the logical flow of the paper.
Sorry for the misinterpretation due to an error. It should be Figure 5 instead of Figure 6, but Figure 5 was substituted with Table 3, and the paragraph next to it was modified.
6. The paper mentions several evaluation metrics, such as mAP50, IoU, precision, and recall, but the specific methods for calculating and applying these metrics in different experiments are not clearly defined. It is recommended that the authors provide further clarification on the calculation process and application of these metrics in each experiment. Additionally, regarding the number of samples with an IoU score below 0.8, the authors should offer more background information and explain the significance and impact of this in the context of the experiments.
We agree. More information on the performance metrics was added at the beginning of Section 3, including reference [31]. An analysis of the performance metrics was added after each experiment.
7. The paper mentions that although all model variants successfully detected faces in thermal images during both training and testing, predictions with IoU scores below 0.8 were still considered acceptable. It is recommended to further analyze these cases with low IoU scores, particularly when faces are at extreme angles or farther from the sensor. Is the model's performance stable in such cases? Are there specific scenarios where the performance significantly declines?
Thank you for your comment. When testing with a subset of the Terravic Facial IR database we found only a very few cases where IoU metric fell below 0.8 and we decided to analyze them using Figure 8 to show that even when these cases presented the lowest IuO values, the difference in position and size of the predicted and annotated bounding boxes does not affect the final goal of detecting faces. We have added new experiments testing the trained models on the Charlotte-ThermalFace database, resulting in lower IoU values. We analyzed the causes of these lower values.
8. Although the performance of the YOLO11 model is presented in this paper, it is recommended that the authors include comparative experiments with traditional methods (such as HOG+SVM, Faster R-CNN, etc.) or other advanced models (such as YOLOv8, YOLOv9, etc.). By comparing the YOLO11 model with these alternatives, its advantages can be highlighted, and its innovation and practicality can be clearly demonstrated.
Thank you for your recommendation. We are now including comparative experiments with YOLOv8 model in section 3.
9. It is recommended to include ablation experiments to validate the actual contribution of the improved backbone/neck structure to the feature extraction from thermal imaging.
Thank you for your recommendation. The Ultralytics package offers methods for data augmentation, including certain forms of ablation that involve removing portions of faces in the images used during training. We have included Figure 5, which displays training batches of images showcasing this type of augmentation.
10. To help readers better understand the role and advantages of the various components of the embedded vision system for thermal face detection, it is recommended that the authors analyze the experimental data in Section 3 and place it immediately after the corresponding performance tables, rather than consolidating all the information in Section 4.
Thank you for your recommendation. We added an analysis of the results after each experiment of Section 3.
11. The embedded vision system for thermal face detection has only been validated and tested on the Terravic Facial IR Database. Does it have practical applicability in an engineering context? The authors are encouraged to validate this through experiments and provide clarification.
Thank you for your comment. We present new experiments testing the YOLOv8 and YOLO11 models in all their variants on the Charlotte-ThermalFace database. And included an analysis of the results.
12. The conclusion section reiterates the authors' contributions. It is recommended that the authors simplify the language and focus on addressing what method was proposed, why it was proposed, what problems it solves, and the potential future developments in this area.
Thank you for your recommendation. The conclusions section was modified to address your comments.
Reviewer 3 Report
Comments and Suggestions for Authors
This paper proposes an embedded vision system based on the YOLO11 model for face detection in thermal infrared images. Thermal infrared sensors have significant advantages, especially in security scenarios in low-light environments. The feasibility of the system was verified by comparing the performance of different model variants and hardware platforms. However, some details need to be further improved. The specific comments are as follows:
- The contribution of the manuscript should be summarized in paragraphs at the end of the introduction. At the same time, a general introduction to the following chapters should be added at the end of the introduction.
- The image in Figure 5 is too narrow and the font is compressed. It is recommended to adjust the length and width of the image.
- The author did not explain why YOLO11 was chosen instead of other models (such as YOLOv8 or EfficientDet).
- The experiment needs to be improved. It is recommended to add a performance comparison between YOLO11 and other models (such as YOLOv8, RetinaNet) on the same dataset.
- I would like to know whether the system can be generalized in other visual tasks (such as autonomous driving, image recognition, etc.). Please cite the references provided and discuss the scalability of the manuscript based on them. [1] Your Data Is Not Perfect: Towards Cross-Domain Out-of-Distribution Detection in Class-Imbalanced Data. [2] Learning discriminative topological structure information representation for 2D shape and social network classification via persistent homology. [3]Pedestrian trajectory prediction in pedestrian-vehicle mixed environments: A systematic review. [4]Trajectory unified transformer for pedestrian trajectory prediction.
Author Response
Thank you for your comments to improve our paper. We addressed your observations as follows:
- The contribution of the manuscript should be summarized in paragraphs at the end of the introduction. At the same time, a general introduction to the following chapters should be added at the end of the introduction.
Thank you for pointing this out. We have modified the contributions summary in the final paragraphs of the introduction section. We have also included a summary of the contents of the subsequent sections at the end of the introduction section.
- The image in Figure 5 is too narrow and the font is compressed. It is recommended to adjust the length and width of the image.
Agree. Figure 5 was replaced with Table 3 to improve the readability of key training hyperparameters, as requested by other reviewers.
- The author did not explain why YOLO11 was chosen instead of other models (such as YOLOv8 or EfficientDet).
Thank you for pointing this out. We started this research working with the YOLOv8 model and moved to the YOLO11 model when it was made available.
- The experiment needs to be improved. It is recommended to add a performance comparison between YOLO11 and other models (such as YOLOv8, RetinaNet) on the same dataset.
Thank you for your recommendation. We present experiments made with the YOLOv8 model and compare results with those of the YOLO11 model.
- I would like to know whether the system can be generalized in other visual tasks (such as autonomous driving, image recognition, etc.). Please cite the references provided and discuss the scalability of the manuscript based on them. [1] Your Data Is Not Perfect: Towards Cross-Domain Out-of-Distribution Detection in Class-Imbalanced Data. [2] Learning discriminative topological structure information representation for 2D shape and social network classification via persistent homology. [3]Pedestrian trajectory prediction in pedestrian-vehicle mixed environments: A systematic review. [4]Trajectory unified transformer for pedestrian trajectory prediction.
Thank you for your suggestion, we included the last two references and discuss the possible application of the proposed system in the context of pedestrian detection for trajectory predictions.
Round 2
Reviewer 1 Report
Comments and Suggestions for Authors
No comments.
Author Response
Thank you again for helping us to improve our paper.Reviewer 2 Report
Comments and Suggestions for Authors
Recommendation: Accept (minor edits)
Comments:
This manuscript describes the development of an embedded vision system for face recognition that aims to detect faces by analyzing images captured by thermal infrared sensors to overcome the limitations imposed by different lighting conditions. The study has some advanced and applied value
However, in its current form, the manuscript contains some shortcomings. Modifications should be made on the following points to argue for a recommendation for publication. During the review process, the following issues were identified:
- In line 93 of the manuscript, “1) Is it,” including the exposition that follows, “Is it” begins a sentence, but there is no punctuation before the sentence.
- In manuscripts 31, 36 and elsewhere, references [1, 2], [3] should precede punctuation marks, and there are numerous instances in the manuscripts where the placement of references is confused.
- YOLOv8” and ‘YOLO11’ appear in the manuscript, and it is suggested that the naming should be standardized by replacing ‘YOLO11’ with ‘YOLOv’. YOLOv11”.
- Why there are no convergence plots for model training in the manuscript, it is suggested to add convergence plots for the model training process.
- Suggest adding experiments and performance graphs to illustrate the theory of the article, not just the more official YOLO model data.

Author Response
Thank you again for helping us to improve our paper. We addressed your observations as follows:
- In line 93 of the manuscript, “1) Is it,” including the exposition that follows, “Is it” begins a sentence, but there is no punctuation before the sentence.
Agree. The numbering method was changed to use the correct punctuation.
- In manuscripts 31, 36 and elsewhere, references [1, 2], [3] should precede punctuation marks, and there are numerous instances in the manuscripts where the placement of references is confused.
Thank you for pointing this out. All the references were placed before the final period of the paragraphs.
- YOLOv8” and ‘YOLO11’ appear in the manuscript, and it is suggested that the naming should be standardized by replacing ‘YOLO11’ with ‘YOLOv’. YOLOv11”.
Thank you for your suggestion. The names published by the developers are used in the paper. After YOLOv10, they are not using the ‘YOLOv’ prefix anymore.
- Why there are no convergence plots for model training in the manuscript, it is suggested to add convergence plots for the model training process.
Thank you for your suggestion. Training and validation loss function plots were added in Figure 7, substituting Table 5.
- Suggest adding experiments and performance graphs to illustrate the theory of the article, not just the more official YOLO model data.
Thank you for your suggestion. Training and validation performance metrics plots were added in Figure 8. We used the training and validation loss functions and performance metrics computed by the Ultralytics package to generate the plots in Matlab.
For the testing experiments, we used the coordinates of the predicted and annotated bounding boxes to compute the performance metrics: Average IoU and Accuracy, with our own written Python scripts.
Reviewer 3 Report
Comments and Suggestions for Authors
The author has addressed most of my questions, but the fifth question was not addressed well. I suggest that the author add [1]Looking Clearer with Text: A Hierarchical Context Blending Network for Occluded Person Re-Identification. [2]Learning discriminative topological structure information representation for 2D shape and social network classification via persistent homology in the final version. If the author can add these two latest works, then I tend to accept the manuscript.
Author Response
Thank you again for helping us to improve our paper. We addressed your observation as follows:
- The author has addressed most of my questions, but the fifth question was not addressed well. I suggest that the author add [1] Looking Clearer with Text: A Hierarchical Context Blending Network for Occluded Person Re-Identification. [2] Learning discriminative topological structure information representation for 2D shape and social network classification via persistent homology in the final version. If the author can add these two latest works, then I tend to accept the manuscript.
Thank you for your suggestion. We included these two references and described their importance in the context of our paper.